

# Gene–environment interaction effect of hypothalamic–pituitary–adrenal axis gene polymorphisms and job stress on the risk of sleep disturbances

Min Zhao[1,*], Yuxi Wang[1,*], Yidan Zeng[1], Huimin Huang[1], Tong Xu[2], Baoying Liu[1], Chuancheng Wu[1], Xiufeng Luo[3] and Yu Jiang[1]

[1] Department of Public Health, Fujian Medical University, Fuzhou, China
[2] Affiliated Zhongshan Hospital of Dalian University, Dalian, China
[3] Fuzhou Municipal Center for Disease Control and Prevention, Fuzhou, China
* These authors contributed equally to this work.

Corresponding author
Yu Jiang, jiangyu@fjmu.edu.cn

## ABSTRACT

**Background:** Studies have shown that chronic exposure to job stress may increase the risk of sleep disturbances and that hypothalamic–pituitary–adrenal (HPA) axis gene polymorphisms may play an important role in the psychopathologic mechanisms of sleep disturbances. However, the interactions among job stress, gene polymorphisms and sleep disturbances have not been examined from the perspective of the HPA axis. This study aimed to know whether job stress is a risk factor for sleep disturbances and to further explore the effect of the HPA axis gene × job stress interaction on sleep disturbances among railway workers.

**Methods:** In this cross-sectional study, 671 participants (363 males and 308 females) from the China Railway Fuzhou Branch were included. Sleep disturbances were evaluated with the Pittsburgh Sleep Quality Index (PSQI), and job stress was measured with the Effort-Reward Imbalance scale (ERI). Generalized multivariate dimensionality reduction (GMDR) models were used to assess gene-environment interactions.

**Results:** We found a significant positive correlation between job stress and sleep disturbances ($P < 0.01$). The *FKBP5* rs1360780-T and rs4713916-A alleles and the *CRHR1* rs110402-G allele were associated with increased sleep disturbance risk, with adjusted ORs (95% CIs) of 1.75 [1.38–2.22], 1.68 [1.30–2.18] and 1.43 [1.09–1.87], respectively. However, the *FKBP5* rs9470080-T allele was a protective factor against sleep disturbances, with an OR (95% CI) of 0.65 [0.51–0.83]. GMDR analysis indicated that under job stress, individuals with the *FKBP5* rs1368780-CT, rs4713916-GG, and rs9470080-CT genotypes and the *CRHR1* rs110402-AA genotype had the greatest risk of sleep disturbances.

**Conclusions:** Individuals carrying risk alleles who experience job stress may be at increased risk of sleep disturbances. These findings may provide new insights into stress-related sleep disturbances in occupational populations.

## INTRODUCTION

Sleep is essential for humans, helping to maintain energy, promote growth and development, and improve immunity (*Ramar et al., 2021*). However, sleep disturbances seriously reduce people's quality of life and have become a major public health problem affecting people's physical and mental health (*Halonen et al., 2017*). The global prevalence of sleep disturbances is approximately 37.9% (*Wu et al., 2021*); in Canada, the prevalence is 23.8% (*Chaput et al., 2018*); in Japan, the prevalence is 13.3% (*Miyachi et al., 2021*); in the US, the prevalence is 30.5% (*Kadier et al., 2023*); in Europe, the prevalence is 25.73% (*Linh et al., 2023*); in Africa, the prevalence is 32.6% (*Wang et al., 2019*); and in China, the prevalence is 29.2% (*Shi et al., 2020*). Long-term sleep disturbances negatively affect people's physical and mental health and are early risk factors for many diseases, such as cardiovascular and cerebrovascular diseases, neuropsychiatric disorders, accidental injuries and even death (*Rajaratnam et al., 2011*; *Morin & Jarrin, 2022*).

Job stress is a negative physical and psychological reaction that occurs when job requirements do not match workers' abilities, coping resources and demands (*Basu, Qayyum & Mason, 2017*). In recent years, studies have shown that excessive job stress can lead to imbalances in physiological functions, resulting in decreased sleep quality and sleep problems such as insomnia and drowsiness (*Khamisa et al., 2016*; *Herr et al., 2018*; *D'Ettorre et al., 2020*). Therefore, job-related stress is a major occupational risk factor that significantly increases the risk of sleep disturbances (*Juster & McEwen, 2015*; *Linton et al., 2015*). Epidemiological research has indicated that job stress is related to an increased risk of sleep disturbances (*Blom et al., 2020*; *Hämmig, 2020*). A prospective cohort study of Japanese workers ($n = 1,022$) with a 2-year observation period also revealed that high job stress was strongly associated with insomnia, with an OR (95% CI) of 1.72 [1.06–2.79] (*Ota et al., 2009*). A prospective cohort study of workers in Denmark aligns with this conclusion (*Nordentoft et al., 2020*). In addition, sleep disturbances also seriously affect the efficiency of workers, leading to a decline in production efficiency and the occurrence of accidents, resulting in substantial social and economic burdens (*Kucharczyk, Morgan & Hall, 2012*; *Uehli et al., 2014*). It is essential to explore the mechanism underlying the influence of job stress on sleep disturbances among occupational groups and to take active measures to reduce the occurrence of sleep disturbances.

The HPA axis is thought to be the main pathway mediating the stress response (*Hirotsu, Tufik & Andersen, 2015*). More importantly, the HPA axis regulates the sleep-wake cycle: activation of the HPA axis may lead to awakening and insomnia in animals and humans (*de Feijter et al., 2022*). Dysfunction of the HPA axis at any molecule (such as the corticotropin-releasing hormone receptor, glucocorticoid receptor or mineralocorticoid receptor) may disturb sleep (*Buckley & Schatzberg, 2005*). When encountering stressors (physiological or psychological), the hypothalamus releases corticotropin-releasing hormone (CRH). CRH stimulates the anterior pituitary to release corticotropin, and corticotropin activates the adrenal cortex to upregulate the production of glucocorticoids (GCs). Its main function is to restore internal physiological balance after exposure to stress. However, *Weitzman et al. (1983)* showed that the release of GCs was

related to the occurrence and development of sleep disturbances. Moreover, most stress-related hormones promote wakefulness, and elevated HPA activity appears to contribute to stress-induced insomnia (*Nicolaides, Vgontzas & Kritikou, 2000*). Exploring the genes that play a role in HPA axis regulation may be useful in determining the relationship between job stress and sleep disturbances. *Gerritsen et al. (2017)* suggested that the *CRH* gene is linked to stress and sleep disturbances. In addition, individual variation in the FK506 binding protein five (*FKBP5*) gene is related to an imbalance in the HPA axis; this imbalance has been identified as the key neurobiological mechanism underlying psychotic symptoms (*Nold et al., 2022*; *Wang et al., 2023*). An animal study also reported that *FKBP5* may be a target gene for stress-induced mood and sleep disturbances (*Albu et al., 2014*). Although many studies have shown that sleep disturbances are related to HPA axis genes and job stress, their interaction and effect on sleep disturbances remain unclear.

In recent years, many researchers have assessed the effects of gene−environment interactions on sleep disturbances (*Zwicker, Denovan-Wright & Uher, 2018*; *Zhang, Khan & Rzhetsky, 2022*). Both genetic (*Federenko et al., 2004*) and environmental factors have been shown to influence an individual's cortisol response to stress through the HPA axis, even if the response is extreme enough to increase the risk of sleep disturbances (*Foley & Kirschbaum, 2010*; *Kudielka & Wüst, 2010*). Moreover, interactions between several genes (the glucocorticoid receptor (*GR*)) (*Bakker et al., 2017*), *FKBP5* (*Matosin, Halldorsdottir & Binder, 2018*; *Normann & Buttenschøn, 2020*), 5-hydroxytryptamine transporter (*5-HTTLPR*) (*Huang et al., 2014*) and dopamine D2 receptor (*DRD2*) (*Jiang et al., 2020*) and exposure to job stress have repeatedly been found to play a role in the onset of sleep disturbances. For instance, *Brummett et al. (2007)* reported that the *5-HTTLPR* gene polymorphism is related to sleep quality problems in individuals exposed to long-term stress. A previous study reported that the effects of early-life stress on mental illnesses such as sleep disturbances were more prominent for the G alleles of the *GR* genes rs258747 and rs41423247 (*Lian et al., 2014*). One of the largest Trier Social Stress Test (TSST) cohorts indicated that interactions among *FKBP5*, corticotrophin-releasing hormone receptor type 1 gene (*CRHR1*) gene polymorphisms and psychosocial stress may affect the cortisol response and cause circadian rhythm disruption (*Mahon et al., 2013*). However, there are still single nucleotide polymorphisms (SNPs) in the HPA axis that have not been fully investigated in these interactions. Most studies have focused on the effect of a single gene-stress interaction on sleep quality, and few have examined multiple major genes regulating the HPA axis to determine the relationships among gene polymorphisms, job stress, and their interaction with sleep disturbances.

Therefore, we chose SNPs of several major genes regulating the HPA axis to investigate the independent and interactive effects of HPA axis gene polymorphisms and job stress on sleep quality among front-line railway workers in Fuzhou city, China. Our investigation focused on the interaction effect of genetic and environmental factors on sleep disturbances to provide new insights for improving sleep health.

## MATERIALS AND METHODS

### Subjects

The present study was conducted as part of the Occupational Health Study for Railway Workers (OHSRW) between October 2019 and May 2020. The inclusion and exclusion criteria have been described in detail in a previous article (*Wang et al., 2022b*). A set of self-report questionnaires was used to collect information on demographic characteristics, sleep disturbances and job stress. As a part of the physical examination, 5-mL fasting venous blood samples were collected from each subject at the workplace between 7:00 am and 9:00 am. In this cross-sectional study, a total of 690 participants were enrolled, 19 of whom were excluded due to insufficient information or missing blood samples. Ultimately, 671 (males/females = 363/308) railway front-line workers were included in the final analysis. This study was approved by the Ethics Committee of Fujian Medical University (No. 2019025). All subjects provided informed consent before they participated in the study and signed a written informed consent form.

### Job stress

The Effort-Reward Imbalance (ERI) scale, which is based on Siegrist's ERI model, was used to evaluate job stress (*Siegrist & Li, 2017*). The Cronbach's alpha of this scale was 0.882. The ERI questionnaire includes a total of 23 items in three dimensions: job effort (six items), job reward (six items) and overcommitment (11 items). Each of the items is evaluated on a five-point scale (from 1 to 5). The ERI score evaluation method is as follows: each item, is assigned the same weight, and the ERI score is calculated as $E/(R \times (6/11))$. ERI scores >1 indicate an imbalance between effort and reward, which is considered to reflect job stress (*Choi et al., 2014*).

### Sleep disturbances

The Pittsburgh Sleep Quality Index (PSQI) was used to assess the sleep quality of the subjects (*Buysse et al., 1989*). The PSQI has shown strong reliability and validity in a variety of samples, indicating that this questionnaire provides a good understanding of sleep disturbances (*Mollayeva et al., 2016*). The PSQI consists of seven components: subjective sleep quality, sleep latency, sleep duration, sleep efficiency, sleep disturbance, sleep medication, and daytime dysfunction. Each dimension is graded on a scale ranging from 0 to 3, and the total PSQI score ranges from 0 to 21. The higher the score is, the worse the sleep quality. It has been reported that the PSQI is an easily accepted and applied tool for assessing sleep disturbances, with a score of ≥5 indicating that the subject has significantly poorer sleep quality. In this study, subjects with a global score higher than five were classified as experiencing sleep disturbances (*Yilmaz, 2020*; *Liu, Kahathuduwa & Vazsonyi, 2021*).

### DNA extraction and genotyping

After a 12-h fast, venous blood samples were collected from all participants using EDTA-containing tubes. Genomic DNA was isolated and purified from the samples using a whole blood genome extraction kit (Beijing Thinkout Sci-Tech Co., Ltd., Beijing, China),

**Table 1 Description of primer sequences.**

| Gene/SNPs | Major/minor alleles | Function | Primer (5′→3′) |
|---|---|---|---|
| *FKBP5* | | | |
| rs1360780 | C/T | Intron variant, risk-factor | Forward: 5′-GGCATGGGCACTCTGAAAAGAT-3′ |
| | | | Reverse: 5′-TCTCTTGTGCCAGCAGTAGCAAGT-3′ |
| rs3800373 | A/C | 3 Prime UTR variant, benign | Forward: 5′-GGCATGGGAAGCTGTCTTCAAC-3′ |
| | | | Reverse: 5′-CCAGCATTGCTACTGCTCAGCTTC-3′ |
| rs9470080 | C/T | Intron variant | Forward: 5′-TCTTTTCCAGGCTATGAATTGACAAA-3′ |
| | | | Reverse: 5′-TGTGTCCAGCCATGTGCTTTTT-3′ |
| rs4713916 | G/A | Intron variant | Forward: 5′-TGGCAACCCTAACCTCTCTGGA-3′ |
| | | | Reverse: 5′-TGTAGGTTCGGGGTACATGTGAAG-3′ |
| rs3777747 | A/G | Intron variant | Forward: 5′-CCGCCTAAGCCTGTTGAGAAGA-3′ |
| | | | Reverse: 5′-TCCAGTTGTTGGCGTACCTCCT-3′ |
| rs9296158 | G/A | Intron variant | Forward: 5′-CACTCGTTCTGTTATACTCATTCCATGC-3′ |
| | | | Reverse: 5′-AGGCCTGGGCTAGGGGTAATTC-3′ |
| *CRHR1* | | | |
| rs110402 | G/A | Intron variant | Forward: 5′-AGATCAGCGGATGGTGAAGAGG-3′ |
| | | | Reverse: 5′-CTTGGCTGCCTAGAACCCTGAC-3′ |
| *CRHR2* | | | |
| rs2267715 | A/G | Intron variant | Forward: 5′-TCTCTCCCAGCAGGGAAGTTGT-3′ |
| | | | Reverse: 5′-CTGGAGGGAGTGGGGGTAAACT-3′ |
| *NR3C1* | | | |
| rs41423247 | G/C | Intron variant | Forward: 5′-GGGGATGAGGTTACGGGGTAGA-3′ |
| | | | Reverse: 5′-TGCTCACAGGGTTCTTGCCATA-3′ |

and the extracted DNA was stored in a −80 °C freezer. Gene polymorphisms were detected by the SNaPshot method (*Larsson et al., 2022*). Tag single nucleotide polymorphisms were derived from a Chinese Han population in the Haplotype Map database (National Center for Biotechnology Information, Bethesda, MD, USA) (*Sayers et al., 2023*). We explored polymorphisms of several major genes that regulate the HPA axis: the *FKBP5* gene (rs1360780, rs3800373, rs9470080, rs4713916, rs3777747, and rs9296158), *CRHR1* gene (rs110402), corticotrophin-releasing hormone type 2 receptor gene (*CRHR2*; rs2267715), and glucocorticoid receptor gene (*NR3C1*; rs41423247). We used SNPs of the above genes, rs1360780 C >T, rs3800373A > C, rs9470080 C > T, rs4713916 G > A, rs3777747 G > A, rs9296158 G > A, rs110402 G > A, rs2267715 G > A and rs41423247 G > C allele combinations, for further haplotype analysis in an attempt to assess the role of haplotypes within the *FKBP5*, *CRHR1*, *CRHR2*, and *NR3C1* genes in susceptibility to sleep disturbances. Table 1 shows the sequences of the primers. The complete sequence is provided in the Supplemental Primer Sequences file.

## Confounding factors

It has been demonstrated that some demographic, socioeconomic and lifestyle factors are related to sleep disturbances; thus, they may influence the results of any

interaction between sleep disturbances and job stress or HPA axis gene polymorphisms (*Wakasugi et al., 2014*). In brief, we included age, sex, ethnicity and marital status as confounding factors. In addition, smoking and drinking alcohol were considered potential confounding lifestyle factors.

## Statistical analysis

Statistical analyses were carried out using SPSS version 26.0 (SPSS Inc., Chicago, IL, USA). The ERI and PSQI scores are presented as the mean ± standard deviation (SD). Differences in demographic data between two groups were compared using the chi-squared test for categorical variables. The Hardy–Weinberg equilibrium (HWE) for the HPA axis gene polymorphisms was tested using a chi-squared goodness-of-fit test. Pearson correlation analysis was used to assess the correlations of job stress with sleep disturbances and job stress dimension scores. After adjusting for sex, age, ethnicity, marital status, smoking status and drinking status as covariates, odds ratios (ORs) and 95% confidence intervals (*Levante et al., 2023*) were determined for the associations of genotypes and job stress with the risk of sleep disturbances by logistic regression. Bonferroni correction was applied to account for multiple comparisons.

The generalized multifactor dimensionality reduction (GMDR) method is a versatile software for detecting gene–gene and gene–environment interactions underlying complex traits (*Xu et al., 2016*). In this study, 0.9 GMDR was used to identify the best HPA axis gene × job stress combination, and we used 10-fold cross-validation and 1,000-fold permutation testing. The GMDR provides numerous output parameters, including cross-validation (CV) consistency, testing balanced accuracy, and empirical *P*-values, to assess each selected interaction. The CV consistency score is a measure of the degree of consistency with which the selected interaction is identified as the best model among all possibilities considered (*Galimova et al., 2017*). We also conducted locus and haplotype analysis for haplotypes associated with sleep disturbances using SHEsis (http://analysis.bio-x.cn/). SHEsis is a powerful software platform for analyses of linkage disequilibrium, haplotype construction, and genetic association at polymorphism loci (*Shi & He, 2005*). Haplotype analysis was performed to indicate the degree of association between alleles of different SNPs, thus assessing the role of common genotypes in susceptibility to sleep disturbances. All reported *P* values are two-tailed, and those less than 0.05 were considered to indicate statistical significance. G Power software showed that the statistical power of this study was 0.73.

## RESULTS

### Demographic characteristics of the subjects

The general demographic characteristics of the patients in the sleep disturbance group and nonsleep disturbance group are summarized in Table 2. A total of 671 subjects were included in this study, including 269 with sleep disturbances and 402 without sleep disturbances. The incidence of sleep disturbances was 40.09%. We found no significant differences between the two groups in terms of sex, age, ethnicity, marital status, smoking status or drinking status ($P > 0.05$). In addition, of the 671 participants, 121 workers (33.1%) were not experiencing job stress but had sleep disturbances, and 148 (48.5%) were

**Table 2 Demographic characteristics of 671 participants in non-sleep disturbance and sleep disturbance group.**

| Variables | N | Non-sleep disturbance (%) | Sleep disturbance (%) | $\chi^2$ | P-value |
|---|---|---|---|---|---|
| Gender | | | | | |
| Male | 363 | 221 (60.9) | 142 (39.1) | 0.31 | 0.58 |
| Female | 308 | 181 (58.8) | 127 (41.2) | | |
| Age (years) | | | | | |
| ≤30 | 159 | 95 (59.7) | 64 (40.3) | 3.97 | 0.27 |
| 31–40 | 234 | 147 (62.8) | 87 (37.2) | | |
| 41–50 | 200 | 109 (54.5) | 91 (45.5) | | |
| >51 | 78 | 51 (65.4) | 28 (34.6) | | |
| Ethnicity | | | | | |
| Han | 526 | 318 (60.5) | 208 (39.5) | 0.30 | 0.58 |
| Minority | 145 | 84 (57.9) | 61 (42.1) | | |
| Marital status | | | | | |
| Unmarried | 118 | 68 (57.6) | 50 (42.4) | 2.04 | 0.36 |
| Married | 517 | 316 (61.1) | 201 (38.9) | | |
| Divorced | 36 | 18 (50.0) | 18 (50.0) | | |
| Smoking status | | | | | |
| Non-smoker | 409 | 246 (60.1) | 163 (39.9) | 0.02 | 0.88 |
| Smoker | 262 | 156 (59.5) | 106 (40.5) | | |
| Alcohol status | | | | | |
| Non-drinker | 310 | 183 (59.0) | 127 (41.0) | 0.19 | 0.67 |
| Drinker | 361 | 219 (60.7) | 142 (39.3) | | |
| Job stress | | | | | |
| Non-job stress | 366 | 245 (66.9) | 121 (33.1) | 16.57 | <0.01 |
| Job stress | 305 | 157 (51.5) | 148 (48.5) | | |

experiencing both job stress and sleep disturbances. There was a significant difference in the distribution of job stress between the two groups ($P < 0.01$).

## Correlation between job stress and sleep disturbances

Table 3 shows the correlations among the ERI scores, PSQI scores, and all dimensions of sleep disturbances. When sex, age, ethnicity, marital status, smoking status and drinking status were controlled as covariates, the ERI score was positively correlated with various dimensions of sleep disturbances, including subjective sleep quality and sleep latency ($P < 0.01$). Specifically, overcommitment showed a meaningful positive correlation with sleep medication and the PSQI, with r values of 0.12 and 0.08, respectively. Job effort showed a meaningful positive correlation with sleep medication (r = 0.11). Importantly, there was a positive correlation between the ERI score and PSQI score (r = 0.16, $P < 0.01$), indicating that job stress is related to sleep disturbances, and the greater the job stress is, the greater the risk of sleep disturbances.

**Table 3 Correlations between the job stress and sleep disturbances and its component scores (n = 671).**

| Variables | Statistical values | Subjective sleep quality | Sleep latency | Sleep duration | Sleep efficiency | Sleep disturbance | Sleep medication | Daytime dysfunction | PSQI |
|---|---|---|---|---|---|---|---|---|---|
| Over-commitment | r | 0.01 | −0.03 | −0.01 | −0.01 | −0.02 | 0.12 | 0.00 | 0.08 |
| | P | 0.86 | 0.45 | 0.88 | 0.76 | 0.62 | **<0.01** | 0.96 | **0.04** |
| Job effort | r | 0.02 | −0.03 | −0.01 | −0.05 | 0.05 | 0.11 | 0.04 | −0.01 |
| | P | 0.61 | 0.45 | 0.74 | 0.21 | 0.22 | **0.01** | 0.26 | 0.84 |
| Job reward | r | 0.05 | −0.02 | 0.00 | 0.01 | −0.01 | 0.05 | −0.02 | 0.01 |
| | P | 0.19 | 0.53 | 0.98 | 0.90 | 0.80 | 0.24 | 0.59 | 0.71 |
| ERI | r | 0.10 | 0.56 | −0.15 | 0.03 | 0.03 | −0.03 | −0.05 | 0.16 |
| | P | **0.01** | **<0.01** | **<0.01** | 0.39 | 0.52 | 0.49 | 0.23 | **<0.01** |

Note:
Adjusted for gender, age, ethnicity, marital status, smoking status and alcohol status; r: correlation coefficient, r < 0 indicates negative correlation, and r > 0 indicates positive correlation. Statistically significant P value was denoted in bold. There were significant positive correlations between ERI and PSQI (r = 0.16, P < 0.01).

## Associations of nine SNPs in HPA axis-related genes with sleep disturbances

The associations of nine SNPs in HPA axis-related genes with sleep disturbances are presented in Table 4. We found that the *FKBP5* rs1360780-TT genotype was associated with increased sleep disturbance risk, with an adjusted OR (95% CI) of 5.34 [3.02–9.44] ($P = 0.001$, Bonferroni-corrected $P < 0.01$). However, the *FKBP5* rs9470080-TT genotype was a protective factor against sleep disturbances, with an adjusted OR (95% CI) of 0.51 [0.28–0.92] ($P = 0.001$, Bonferroni-corrected $P < 0.01$). The *FKBP5* rs1360780-T and rs4713916-A alleles and the *CRHR1* rs110402-G allele were risk factors for sleep disturbances, with adjusted ORs (95% CIs) of 1.75 [1.38–2.22], 1.68 [1.30–2.18] and 1.43 [1.09–1.87], respectively (all $P = 0.001$, Bonferroni-corrected $P < 0.01$). However, the *FKBP5* rs9470080-T allele was a protective factor against sleep disturbances, with an OR (95% CI) of 0.65 [0.51–0.83] ($P = 0.001$, Bonferroni-corrected $P < 0.01$). Haplotype analysis revealed significant differences in the haplotypes between the sleep disturbance group and the nonsleep disturbance group. The C-A-G-A-G-C haplotype was associated with an increased risk of sleep disturbances, and details are provided in the Supplemental File (Table S1).

## Effect of the gene–environment interaction on sleep disturbances

When sex, age, ethnicity, marital status, smoking status and drinking status were controlled as covariates, the best gene–environment interaction models were determined by GMDR analysis (Table 5). These models showed a significant effect of the interaction between HPA axis genes and job stress on sleep disturbances. The model had the maximum cross-validation consistency coefficient (10/10), and the accuracies of the training set and testing set were 0.68 and 0.60, respectively. ERI × rs1360780 × rs947008 × rs4713916 × rs110402 was considered the best interaction model because it contained the most SNPs among the models that met the best model criteria. This suggests that the best interaction model was the interaction between job stress and *FKBP5* rs1360780, rs9470080, and rs4713916 genotypes and the *CRHR1* rs110402 genotype. Furthermore, we also found

**Table 4 Associations of nine SNPs in HPA axis related genes with sleep disturbances.**

| Genes | SNPs | Genotypes & alleles | Frequencies N (%) | | OR (95% CI) | HWE | |
|---|---|---|---|---|---|---|---|
| | | | Non-sleep disturbance (n = 402) | Sleep disturbance (n = 269) | | Non-sleep disturbance | Sleep disturbance |
| FKBP5 | rs1360780 | CC | 231 (57.5) | 123 (45.7) | 1.00 | 0.88 | 0.09 |
| | | CT | 152 (37.8) | 93 (34.6) | 1.15 [0.82–1.61] | | |
| | | TT | 19 (4.7) | 53 (19.7) | 5.24 [2.97–9.24]* | | |
| | | C allele | 614 (76.4) | 339 (63.0) | 1.00 | | |
| | | T allele | 190 (23.6) | 199 (37.0) | 1.75 [1.38–2.22]* | | |
| | rs3800373 | AA | 234 (58.2) | 156 (58.0) | 1.00 | 0.90 | 0.71 |
| | | AC | 149 (37.1) | 91 (33.8) | 0.92 [0.66–1.28] | | |
| | | CC | 19 (4.7) | 22 (8.2) | 1.74 [0.91–3.32] | | |
| | | A allele | 617 (76.7) | 403 (74.9) | 1.00 | | |
| | | C allele | 187 (23.2) | 135 (25.1) | 1.11 [0.86–1.43] | | |
| | rs9470080 | CC | 187 (46.5) | 140 (52.0) | 1.00 | 0.92 | 0.90 |
| | | CT | 170 (42.3) | 112 (41.6) | 0.88 [0.64–1.22] | | |
| | | TT | 45 (11.2) | 17 (6.3) | 0.51 [0.28–0.92]* | | |
| | | C allele | 544 (67.7) | 392 (72.9) | 1.00 | | |
| | | T allele | 260 (32.3) | 146 (27.1) | 0.65 [0.51–0.83]* | | |
| | rs4713916 | GG | 256 (63.7) | 152 (56.5) | 1.00 | 0.99 | 0.99 |
| | | GA | 130 (32.3) | 99 (36.8) | 1.28 [0.92–1.79] | | |
| | | AA | 16 (4.0) | 18 (6.7) | 1.90 (0.94–3.83) | | |
| | | G allele | 642 (79.9) | 403 (74.9) | 1.00 | | |
| | | A allele | 162 (20.1) | 135 (25.1) | 1.68 [1.30–2.18]* | | |
| | rs3777747 | AA | 66 (16.4) | 50 (18.6) | 1.00 | 0.77 | 0.81 |
| | | GA | 179 (44.5) | 120 (44.6) | 0.89 [0.58–1.37] | | |
| | | GG | 157 (39.1) | 99 (36.8) | 0.83 [0.53–1.30] | | |
| | | A allele | 515 (64.1) | 220 (40.9) | 1.00 | | |
| | | G allele | 289 (35.9) | 318 (59.1) | 1.13 [0.90–1.41] | | |
| | rs9296158 | GG | 202 (50.2) | 131 (48.7) | 1.00 | 1.00 | 0.87 |
| | | GA | 167 (41.5) | 109 (40.5) | 1.01 [0.72–1.40] | | |
| | | AA | 33 (8.2) | 29 (10.8) | 1.36 [0.79–2.34] | | |
| | | G allele | 571 (71.0) | 371 (69.0) | 1.00 | | |
| | | A allele | 233 (29.0) | 167 (31) | 1.13 [0.89–1.43] | | |
| CRHR1 | rs110402 | AA | 316 (78.6) | 201 (74.7) | 1.00 | 0.73 | 0.43 |
| | | GA | 78 (19.4) | 57 (21.2) | 1.15 [0.78–1.69] | | |
| | | GG | 8 (2.0) | 11 (4.1) | 2.16 [0.86–5.47] | | |
| | | A allele | 94 (11.7) | 79 (14.7) | 1.00 | | |
| | | G allele | 710 (88.3) | 459 (85.3) | 1.43 [1.09–1.87]* | | |
| CRHR2 | rs2267715 | AA | 79 (59.5) | 61 (22.7) | 1.00 | 0.69 | 0.89 |
| | | GA | 183 (34.3) | 127 (47.2) | 0.90 [0.60–1.35] | | |
| | | GG | 140 (6.2) | 81 (30.1) | 0.75 [0.49–1.15] | | |
| | | A allele | 341 (42.4) | 249 (46.3) | 1.00 | | |
| | | G allele | 463 (57.6) | 289 (53.7) | 0.93 [0.74–1.15] | | |

| Genes | SNPs | Genotypes & alleles | Frequencies N (%) | | OR (95% CI) | HWE | |
|---|---|---|---|---|---|---|---|
| | | | Non-sleep disturbance (*n* = 402) | Sleep disturbance (*n* = 269) | | Non-sleep disturbance | Sleep disturbance |
| *NR3C1* | rs41423247 | GG | 258 (64.2) | 168 (62.5) | 1.00 | 0.65 | 0.81 |
| | | GC | 122 (30.3) | 85 (31.6) | 1.07 [0.76–1.50] | | |
| | | CC | 22 (5.5) | 16 (5.9) | 1.12 [0.57–2.19] | | |
| | | G allele | 638 (79.4) | 421 (78.3) | 1.00 | | |
| | | C allele | 166 (20.6) | 117 (21.7) | 1.10 [0.84–1.43] | | |

**Notes:**
Adjusted for gender, age, ethnicity, marital status, smoking status, and alcohol status; the chi-square goodness-of-fit test showed that the genotypic frequencies of nine SNPs in HPA axis related genes in the Non-sleep disturbance group and the sleep disturbance group were consistent with Hardy-Weinberg equilibrium ($P > 0.05$).
* $P < 0.01$.

**Table 5 Best gene-environment interaction models, as identified by GMDR.**

| Model | Training accuracy (%) | Testing accuracy (%) | Cross-validation consistency | *P*-value |
|---|---|---|---|---|
| ERI | 0.58 | 0.56 | 8/10 | 0.17 |
| ERI × rs1360780 | 0.62 | 0.61 | 10/10 | **0.01** |
| ERI × rs1360780 × rs947008 | 0.64 | 0.60 | 4/10 | **<0.01** |
| ERI × rs1360780 × rs947008 × rs110402 | 0.66 | 0.64 | 10/10 | **<0.01** |
| ERI × rs1360780 × rs947008 × rs4713916 × rs110402 | 0.68 | 0.60 | 10/10 | **<0.01** |

**Note:**
Adjusted for gender, age, ethnicity, marital status, smoking status and alcohol status. The best interaction model was selected based on the balance test error of the 1/10 test sample, the accuracy of the cross-validation and *P*-value and more SNPs included in the model, suggest that ERI × rs1360780 × rs947008 × rs4713916 × rs110402 is the best interaction model (Cross-Validation Consistency:10/10, $P < 0.01$). Statistically significant *P* value was denoted in bold.

that under job stress, the subjects with *FKBP5* rs1368780-CT, rs4713916-GG, and rs9470080-CT genotypes and the *CRHR1* rs110402-AA genotype had the greatest risk of sleep disturbances (Fig. 1). In addition, we analyzed the HPA axis gene–gene interactions, and the results showed that rs1360780 × rs947008 × rs110402 was the best gene–gene interaction model among the nine SNPs in the genes related to the HPA axis (Table S2 and Fig. S1).

## DISCUSSION

To our knowledge, this is the first study to investigate the associations among multiple HPA axis gene polymorphisms, job stress, and their interactions with sleep disturbances. Our study has three main findings. (a) After controlling for confounding factors such as sex, age and ethnicity, job stress was correlated with sleep disturbances. (b) *FKBP5* rs1360780-T and rs4713916-A alleles and the *CRHR1* rs110402-G allele were associated with the risk of sleep disturbances. However, the *FKBP5* rs9470080-T allele was a protective factor against sleep disturbances. (c) GMDR analysis showed that in individuals under job stress, the risk of sleep disturbances was the highest for the *FKBP5* rs1368780-CT, rs4713916-GG, and rs9470080-CT genotypes and the *CRHR1* rs110402-AA genotype.

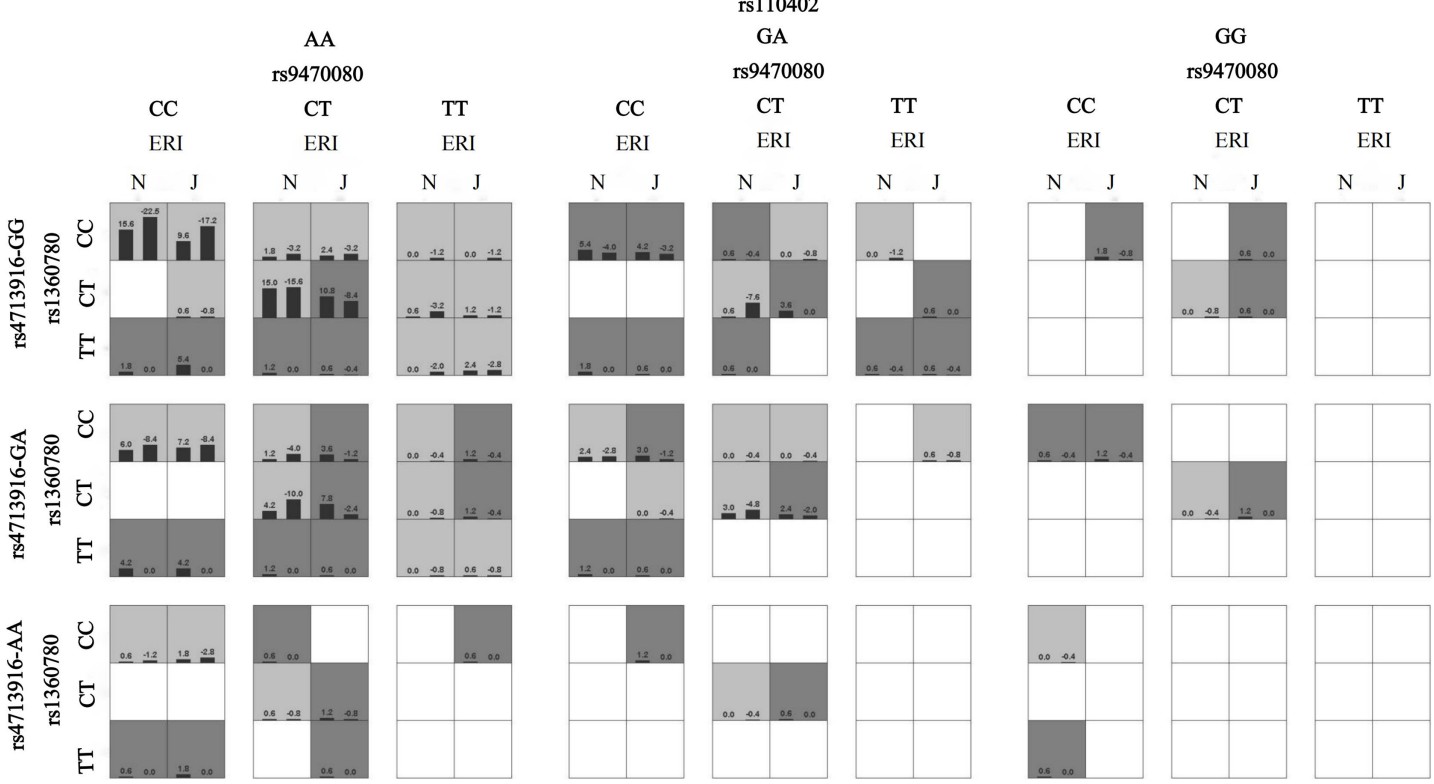

**Figure 1 The interaction model between ERI and HPA genes on sleep disturbances.** A box represents an interaction combination, the darker the color of the box, the higher the risk of the combination. Bars represent the maximum likelihood estimation of case weights. In the same box, the left column is the positive score of the combination, and the right is the negative score; the higher the positive score, the higher the risk of the combination. In the present study, the dark gray box represents the high sleep disturbances risk factors, and the light gray represents the low sleep disturbances risk factors. N and J denote normal and job stress (ERI > 1), respectively. The best gene-environment interaction model is shown in (the third and fourth columns of the second row). Under the job stress (J) (the fourth column of the second row), the rs1368780-CT, rs4713916-GG, and rs9470080-CT and the rs110402-AA interacted with the highest scores, and the positive scores were greater than the negative scores (10.8, −8.4), which indicates that under job stress, the subjects with the *FKBP5* rs1368780-CT, rs4713916-GG, and rs9470080-CT genotypes and the *CRHR1* rs110402-AA genotype had the highest sleep disturbance risk.

In this study, we found that the greater the level of job stress experienced, the worse the sleep quality. Consistent with previous studies, many studies have shown that high job stress is associated with a greater risk of insomnia (*Deguchi et al., 2017*; *Yang et al., 2018*; *Wang et al., 2022a*). In addition, overcommitment was also meaningfully and positively correlated with the PSQI score. This result also suggested that overcommitment and job stress may be related to sleep disturbances (*Yoshioka et al., 2013*; *Wang et al., 2020*). Lallukka et al. reached the same conclusion (*Lallukka et al., 2014*). Job stress is a very influential environmental factor for sleep (*Gosling et al., 2014*). There is evidence that the basal levels of cortisol are elevated in individuals experiencing job stress, and the HPA axis of people experiencing job stress may release the cortisol that causes sleep disturbances (*Fogelman & Canli, 2018*; *Rotvig et al., 2019*). In addition, *Birch & Vanderheyden (2022)* explored how job stress mediates stress-induced insomnia by regulating the glucocorticoid signaling pathway in brain astrocytes. This evidence suggests that job stress interferes with normal sleep and even increases the risk of sleep disturbances by activating the HPA axis.

Consistent with previous results, our study also revealed correlations between several major HPA axis regulatory genes and sleep disturbances. This result indicated that individuals with *FKBP5* rs1360780-T and rs4713916-A alleles and the *CRHR1* rs110402-G allele had a greater risk of sleep disturbance. This finding is in line with a study by *White et al. (2012)* that showed that the interaction between *FKBP5* minor alleles (including rs1360780-T and rs4713916-A alleles) and emotional neglect may increase the risk of stress-related disorders such as sleep disturbances. In addition, previous studies have shown that participants with the *CRHR1* rs110402-A allele had higher cortisol levels 15 min poststress, implying a risk of sleep disturbances in the future (*Weeger et al., 2020*; *Nold et al., 2021*). A meta-analysis showed that individuals exposed to stress and carrying the rs1360780-T allele and rs3800373-C allele had significantly shorter sleep durations and greater risks of stress-related diseases (*Wang, Shelton & Dwivedi, 2018*). Moreover, a study by *Maguire et al. (2020)* suggested that stress-related alterations in HPA axis genes in individuals with PTSD may contribute to sleep difficulties. We also found a protective effect of the *FKBP5* rs9470080-TT genotype against sleep disturbances, which is different from the results of another study (*Li et al., 2019*). We believe that these inconsistent results may be caused by different types of stress or stressors. Previous findings have been based primarily on posttraumatic stress in earthquake survivors. Another possible explanation for the differences is the questionnaires and evaluation criteria used.

Our findings provide new insights into the effects of gene–environment interactions on sleep disturbances. We found that the HPA axis gene × job stress interaction strongly affects sleep disturbances. More importantly, the GMDR results showed that individuals with the *FKBP5* rs1360780-CC genotype, rs9470080-CC genotype and *CRHR1* rs110402-AA genotype have the highest risk of sleep disturbances under job stress. Previous studies have also revealed effects of gene–environment interactions on sleep disturbances. For example, *Zimmermann et al. (2011)* reported that individuals carrying risk alleles of two *FKBP5* SNPs (rs3000377 and rs47139611) have the highest risk of reduced sleep quality if they have experienced adverse life events. Similar results were found for the interaction between childhood trauma and risk alleles of these SNPs (*Bevilacqua et al., 2012*). Similarly, *He et al. (2019)* investigated 712 participants in a large general hospital in Beijing, and the results suggested that when experiencing work-related stress, individuals with the *CRHR1* rs110402-A allele may experience reduced sleep quality. In summary, our study provides evidence that the HPA axis gene × job stress interaction may play an important role in sleep disturbances. Furthermore, according to previous research, the gene × stress interaction can be explained by the diathesis-stress model (*Belsky & Pluess, 2009*). The model suggests that individuals with "risk-associated genes" are prone to stress-related diseases such as sleep disturbances when confronted with stress or adverse environments, while individuals with "resilient-associated genes" are not affected (*Monroe & Simons, 1991*; *Shao et al., 2018*). As diathesis-stress research has highlighted, the interaction of *FKBP5* variants with trauma and adverse environments has been found to confer risk for several psychopathological phenotypes (*Zannas et al., 2016*). In this study, the *FKBP5* rs1360780-CC and rs9470080-CC genotypes and the *CRHR1* rs110402-AA genotype may be risk factors for susceptibility to stressful environments, supporting the

diathesis-stress model. Therefore, to reduce the risk of sleep disturbances, individuals with genetic susceptibility should avoid or reduce job stress as much as possible.

This study has several strengths. This is the first study to examine the effects of multiple gene polymorphisms and job stress on sleep disturbances from the perspective of the HPA axis and to determine a haplotype that increases the power to detect genetic associations (*Aziz et al., 2021*). Haplotype analysis can assess the role of different genotypes of the target gene in susceptibility to sleep disturbances. Furthermore, GMDR was used to investigate the pattern of gene × environment interactions, as it recognizes interactions between multiple loci or environmental factors (*Hou et al., 2019*). However, this research still has some limitations that can be addressed in future studies. First, the evaluation of sleep disturbances was entirely based on the PSQI, and the evaluation of job stress was based exclusively on the ERI, which are subjective questionnaires that are prone to produce false positive results, which may have affected the accuracy of the results. Second, there are different sources of sample bias, including reaction bias (*e.g.*, subjects with poor sleep quality may be more inclined to complete the study than those with good sleep quality) and sample-selection bias (*e.g.*, first-line railway workers are apt to work long hours in stressful environments). Finally, a cross-sectional design was used; thus, we could not examine the causality of the HPA axis gene × job stress interaction in the development of sleep disturbances. In future research, longitudinal designs should be used to further study this causal relationship, as well as experimental methods to measure subjects' sleep disturbances and job stress to provide experimental support for the results of the present study. Importantly, we will incorporate transcriptomic, epigenomic, or HPA axis activity data to conduct more in-depth studies. The results of this study suggest that individuals who carry risk alleles may be at increased risk for sleep disturbances if they are under job stress. This study provides a reliable basis for formulating strategies to reduce employees' job stress and improve sleep quality. This study suggested that industries should pay attention to the occupationally stressful situations of their workers by reducing the incidence of occupational stress (reduction of working hours and tasks, work incentives and support) and thus reducing the incidence of sleep disturbances.

## CONCLUSIONS

This is the first study to investigate the effect of the interaction between job stress and HPA axis gene polymorphisms on sleep disturbances in railway frontline workers. The present study revealed that 48.5% of workers experienced both job stress and sleep disturbances. As the main effect of sleep quality, job stress was found to increase the risk of sleep disturbances. The *FKBP5* rs1360780-T and rs4713916-A alleles and the *CRHR1* rs110402-G allele were also risk factors for sleep disturbances. More importantly, the GMDR results showed that the interactions of SNPs with job stress increased the risk of sleep disturbances, which is the core conclusion of our study. These findings provide new insight into the correlation between job stress and HPA axis gene polymorphisms and their interaction with sleep disturbances.

## ACKNOWLEDGEMENTS

The authors would like to express their sincere gratitude to all participants who participated in the study.

### Funding

This study was supported by the Fujian Medical University's Research Foundation for Talented Scholars (grant number XRCZX2018011) and the Fuzhou Science and Technology Project (grant number 2022-S-033). The funders had no role in study design, data collection and analysis, decision to publish, or preparation of the manuscript.

### Grant Disclosures

The following grant information was disclosed by the authors:
Fujian Medical University's Research: XRCZX2018011.
Fuzhou Science and Technology Project: 2022-S-033.

### Competing Interests

The authors declare that they have no competing interests.

### Author Contributions

- Min Zhao performed the experiments, analyzed the data, authored or reviewed drafts of the article, and approved the final draft.
- Yuxi Wang performed the experiments, analyzed the data, authored or reviewed drafts of the article, and approved the final draft.
- Yidan Zeng performed the experiments, prepared figures and/or tables, and approved the final draft.
- Huimin Huang performed the experiments, prepared figures and/or tables, and approved the final draft.
- Tong Xu performed the experiments, prepared figures and/or tables, and approved the final draft.
- Baoying Liu performed the experiments, prepared figures and/or tables, and approved the final draft.
- Chuancheng Wu performed the experiments, prepared figures and/or tables, and approved the final draft.
- Xiufeng Luo performed the experiments, prepared figures and/or tables, and approved the final draft.
- Yu Jiang conceived and designed the experiments, authored or reviewed drafts of the article, project administration, and approved the final draft.

### Human Ethics

The following information was supplied relating to ethical approvals (*i.e.*, approving body and any reference numbers):

The study was conducted in accordance with the Declaration of Helsinki, and the protocol was approved by the institutional ethical committees of Fujian Medica University (No. 2019025).

## Data Availability
The raw measurements are available in the Supplemental File.

## Supplemental Information
Supplemental information for this article can be found online at http://dx.doi.org/10.7717/peerj.17119#supplemental-information.

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
