# Peer review of "Gene‒environment interaction effect of hypothalamic‒pituitary‒adrenal axis gene polymorphisms and job stress on the risk of sleep disturbances"

_PeerJ, doi:10.7717/peerj.17119_

## Round 0.1 · original submission · Minor Revisions

Please attend carefully to the comments expressed by the reviewers and submit the corresponding new version, including the detailed response.

·

Basic reporting

No comment

Experimental design

Sleep quality variable is dichotomized with a threshold of 5 to consider patients as experiencing sleep disturbance. This decision is not explained, since the PSQI would yield a continuous variable and linear regressions could be performed which might yield more interesting information. At least it should be explained why this methodological decision was taken and how was the threshold of 5 established.

Regarding the haplotype analysis, there is no information on how were haplotype blocks selected.

Validity of the findings

I find the results are interesting, well supported by the data and statistically sound. My main concern would be again related to the categorization of sleep disturbance when PSQI is higher than 5 which yields a rather high prevalence of sleep disturbance in the sample (> 40%).

Finally, within the limitations section, it is stated that the PSQI is a subjective questionnaire and therefore prone to bias. In my understanding this limitation extends to the ERI scale and this should be stated.

Also it might be interesting to include in the final parts of the discussion, the authors opinion as to what might be the most plausible underlying mechanism linking HPA axis genetic variations and sleep disturbances (differential activity of the HPA axis, interactions with clock genes...). This work doesn't count with any transcriptomic, epigenomic or HPA axis activity data and future studies including these variables might help understand this pathway and the authors opinion on future directions would be interesting.

Additional comments

This work addresses an interesting topic; the methods are sound and results provide valuable new findings in this field. However, a few minor issues should be addressed prior to publication.

Reviewer 2 ·

Basic reporting

The manuscript is well written, contains enough and pertinent literature references. The structure of the article is professional.
Figure 1 should be better explained in order the readers can interpret their results of gene-environment interactions.

Experimental design

The goal of the manuscript is clear in order to answer the relationship between HPA axis, job stress and sleep disturbances.
I suggest to explain in a more detailed manner the MDR figure and results.

Validity of the findings

Authors describe a genetic association in a transversal and observational study. Therefore, authors should be cautious with the discussion, this findings must be validated in future studies in a biological context. Authors should include the statistical power for this study.

Additional comments

Peer J
This cross-sectional study involved 671 railway workers from China to explore a potential relationship between HPA gene variants and sleep disorders. Authors concluded that carriers of the risk alleles experiencing job stress may be at increased risk of sleep disturbances.
Comments to the authors:
-The Pittsburgh Sleep Quality Index (PSQI) evaluates sleep quality, and does not evaluate all sleep disturbances. PSQI as a self-rated questionnaire it is a subjective measurement of sleep quality of the participants. Authors did not use objective measures to evaluate sleep disturbances (i.e., polysomnography and actigraphy).

-ERI scale is also a self-reported, I mean authors did not measure any stress-associated biological marker (e.g., cortisol levels as the main glucocorticoid, adrenocorticotropic hormone (ACTH)). In this context, the present work only presents correlations, further studies are required to validate these associations. Please, discuss it and comment on these limitations.

-In Table 3 “Correlations between the job stress and sleep disturbance and its component scores (n= 671)” I see some significant, but very weak correlations (between 0.08 to 0.16). This should be commented in the Discussion section.

-Please, include the functional effect of each variant explored. Correct the following typo in table 1: “Forward:52-5CAC […]”
Also, in Table 1, it would be important to include the “extension primer” for the snapshot technique. So, other researchers may replicate this methodology.

-For a better comprehension of the results, I suggest to include (it could be a supplementary figure) a gene-gene interaction map using MDR. In order to explore whether there is a synergistic or non-additive relationship between the genes analyzed.
-Additionally, did the authors explore possible protein-protein interactions between the gene products studied by pathway enrichment (in HPA axis, or sleep disturbances or psychological stress)?

-Please, include the statistical power calculation for your study. I would like to see the effect size (i.e., the contribution of the SNPs to the genetic variance of the trait analyzed).

-I guess railway workers have long work hours and this justify the job stress and sleep disturbances mentioned in the text. However, participants aged between 20 and 60 years. Authors must consider that sleep disturbances increase in older people. Did the authors stratify in young vs senior workers?

Minor comments:
-Please, define abbreviations when you use them at first time: SNPs (line 114).
-Gene symbols should be italicized throughout the manuscript.
-In the footnote of tables 3, 4 and 5 are captions “a, b and c” for an explanation in each table, however, they are not shown in the Table.
-Please, double check for typos in the entire manuscript (line 257: thatr)

·

Basic reporting

please add literature on depression-FKBP5-sleep disturbances

Experimental design

statistical methods (GMDR) and haplotype analysis are not sufficiently explained.

Why was this way of model selection performed? What were the (a priori defined?) cut-off criteria for choosing the 5-way interaction model over the 4-way interaction model?

Versioning of the code for GMDR is not ensured but would be needed to reproduce the findings upon re-analysis.

Please explain better why (only) the investigated SNPs were chosen (and not others being present in HPA axis-regulating genes)

Validity of the findings

the conclusion on how the findings might be used to remedy sleep disturbances appear far fetched.

Please explain better why the finding on the protective effect of one FKBP5 SNP differs from the provided literature (given reason is rather generic)

was the split of data for cross validation performed randomly or (recommended) stratified based on covariates?

Additional comments

the figure presenting the main result of the study is hard to read / understand. Please revise and add uncertainty information for the provided estimates.

---

## Round 0.2 · Minor Revisions

Please attend to the Reviewer 2 comments. Particularly checking the entire revised manuscript for English style usage.

**Language Note:** The Academic Editor has identified that the English language must be improved. PeerJ can provide language editing services - please contact us at [email protected] for pricing (be sure to provide your manuscript number and title). Alternatively, you should make your own arrangements to improve the language quality and provide details in your response letter. – PeerJ Staff

·

Basic reporting

No comment

Experimental design

No comment

Validity of the findings

No comment

Additional comments

No comment

Reviewer 2 ·

Basic reporting

I highly recommend checking the entire revised manuscript for English style usage, as I still found several typos in the text.

Experimental design

MDR method is better explained in this revised version of the manuscript.
-In statistical analysis section (line 222), I suggest to include the meaning of MDR, as follows: “Generalized multifactor dimensionality reduction (GMDR) method is a Versatile Software for […]”

Validity of the findings

Authors have included references related to FKBP5 gene and depression and discussed their contradictory findings regarding this gene. They also have improved the discussion and conclusions.

Additional comments

The authors answered all my questions and, in general, I see that they addressed all the reviewers' comments. Overall, the manuscript was improved.

Below, you will find minor additional comments to this revised version:
-Genic downstream... “in table 1” please, replace by “3 Prime UTR Variant”
-Please, correct Widowed in “data introduction file”
-Line 60: I suggest Africa as a continent instead of “African”.

I highly recommend checking the entire revised manuscript for English style usage, as I still found several typos in the text.

---

## Round 0.3 · accepted · Accept

Thank you for the revised version of your manuscript, in which I can see that you have addressed all of the reviewers' comments. I acknowledge your improved manuscript. In my opinion, this version is ready for publication.